# Robust Spectral Detection of Global Structures in the Data by Learning a Regularization

**Pan Zhang**
Institute of Theoretical Physics, Chinese Academy of Sciences, Beijing 100190, China
panzhang@itp.ac.cn

## Abstract

Spectral methods are popular in detecting global structures in the given data that can be represented as a matrix. However when the data matrix is sparse or noisy, classic spectral methods usually fail to work, due to localization of eigenvectors (or singular vectors) induced by the sparsity or noise. In this work, we propose a general method to solve the localization problem by learning a regularization matrix from the localized eigenvectors. Using matrix perturbation analysis, we demonstrate that the learned regularizations suppress down the eigenvalues associated with localized eigenvectors and enable us to recover the informative eigenvectors representing the global structure. We show applications of our method in several inference problems: community detection in networks, clustering from pairwise similarities, rank estimation and matrix completion problems. Using extensive experiments, we illustrate that our method solves the localization problem and works down to the theoretical detectability limits in different kinds of synthetic data. This is in contrast with existing spectral algorithms based on data matrix, non-backtracking matrix, Laplacians and those with rank-one regularizations, which perform poorly in the sparse case with noise.

## 1 Introduction

In many statistical inference problems, the task is to detect, from given data, a global structure such as low-rank structure or clustering. The task is usually hard to solve since modern datasets usually have a large dimensionality. When the dataset can be represented as a matrix, spectral methods are popular as it gives a natural way to reduce the dimensionality of data using eigenvectors or singular vectors. In the point-of-view of inference, data can be seen as measurements to the underlying structure. Thus more data gives more precise information about the underlying structure.

However in many situations when we do not have enough measurements, i.e. the data matrix is sparse, standard spectral methods usually have localization problems thus do not work well. One example is the community detection in sparse networks, where the task is to partition nodes into groups such that there are many edges connecting nodes within the same group and comparatively few edges connecting nodes in different groups. It is well known that when the graph has a large connectivity $c$, simply using the first few eigenvectors of the adjacency matrix $A \in \{0,1\}^{n \times n}$ (with $A_{ij} = 1$ denoting an edge between node $i$ and node $j$, and $A_{ij} = 0$ otherwise) gives a good result. In this case, like that of a sufficiently dense Erdős-Rényi (ER) random graph with average degree $c$, the spectral density follows Wigner's semicircle rule, $P(\lambda) = \sqrt{4c - \lambda^2}/2\pi c$, and there is a gap between the edge of bulk of eigenvalues and the informative eigenvalue that represents the underlying community structure. However when the network is large and sparse, the spectral density of the adjacency matrix deviates from the semicircle, the informative eigenvalue is hidden in the bulk of eigenvalues, as displayed in Fig. 1 left. Its eigenvectors associated with largest eigenvalues (which are roughly proportional to $\log n / \log \log n$ for ER random graphs) are localized on the large-

degree nodes, thus reveal only local structures about large degrees rather than the underlying global structure. Other standard matrices for spectral clustering [19, 22], e.g. Laplacian, random walk matrix, normalized Laplacian, all have localization problems but on different local structures such as dangling trees.

Another example is the matrix completion problem which asks to infer missing entries of matrix $A \in \mathbb{R}^{m \times n}$ with rank $r \ll \sqrt{mn}$ from only few observed entries. A popular method for this problem is based on the singular value decomposition (SVD) of the data matrix. However it is well known that when the matrix is sparse, SVD-based method performs very poorly, because the singular vectors corresponding to the largest singular values are localized, i.e. highly concentrated on high-weight column or row indices.

A simple way to ease the pain of localization induced by high degree or weight is trimming [6, 13] which sets to zero columns or rows with a large degree or weight. However trimming throws away part of the information, thus does not work all the way down to the theoretical limit in the community detection problem [6, 15]. It also performs worse than other methods in matrix completion problem [25].

In recent years, many methods have been proposed for the sparsity-problem. One kind of methods use new linear operators related to the belief propagation and Bethe free energy, such as the non-backtracking matrix [15] and Bethe Hessian [24]. Another kind of methods add to the data matrix or its variance a rank-one regularization matrix [2, 11, 16–18, 23]. These methods are quite successful in some inference problems in the sparse regime. However in our understanding none of them works in a general way to solve the localization problem. For instance, the non-backtracking matrix and the Bethe Hessian work very well when the graph has a locally-tree-like structure, but they have again the localization problems when the system has short loops or sub-structures like triangles and cliques. Moreover its performance is sensitive to the noise in the data [10]. Rank-one regularizations have been used for a long time in practice, the most famous example is the "teleportation" term in the Google matrix. However there is no satisfactory way to determine the optimal amount of regularization in general. Moreover, analogous to the non-backtracking matrix and Bethe Hessian, the rank-one regularization approach is also sensitive to the noise, as we will show in the paper.

The main contribution of this paper is to illustrate how to solve the localization problem of spectral methods for general inference problems in sparse regime and with noise, by learning a proper regularization that is specific for the given data matrix from its localized eigenvectors. In the following text we will first discuss in Sec. 2 that all three methods for community detection in sparse graphs can be put into the framework of regularization. Thus the drawbacks of existing methods can be seen as improper choices of regularizations. In Sec. 3 we investigate how to choose a good regularization that is dedicated for the given data, rather than taking a fixed-form regularization as in the existing approaches. We use matrix perturbation analysis to illustrate how the regularization works in penalizing the localized eigenvectors, and making the informative eigenvectors that correlate with the global structure float to the top positions in spectrum. In Sec. 4 we use extensive numerical experiments to validate our approach on several well-studied inference problems, including the community detection in sparse graphs, clustering from sparse pairwise entries, rank estimation and matrix completion from few entries.

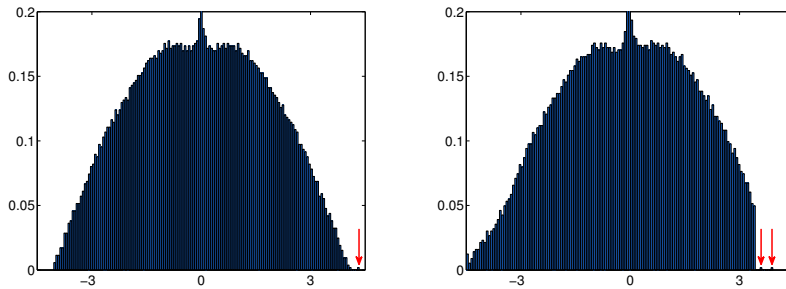

Figure 1: Spectral density of the adjacency matrix (*left*) and X-Laplacian (*right*) of a graph generated by the stochastic block model with $n = 10000$ nodes, average degree $c = 3$, $q = 2$ groups and $\epsilon = 0.125$. Red arrows point to eigenvalues out of the bulk.

## 2 Regularization as a unified framework

We see that the above three methods for the community detection problem in sparse graphs, i.e. trimming, non-backtracking/Bethe Hessian, and rank-one regularizations, can be understood as doing different ways of regularizations. In this framework, we consider a regularized matrix

$$L = \hat{A} + \hat{R}. \tag{1}$$

Here matrix $\hat{A}$ is the data matrix or its (symmetric) variance, such as $\tilde{A} = D^{-1/2}AD^{-1/2}$ with $D$ denoting the diagonal matrix of degrees, and matrix $\hat{R}$ is a regularization matrix. The rank-one regularization approaches [2, 11, 16–18, 23] fall naturally into this framework as they set $R$ to be a rank-one matrix, $-\zeta \mathbf{1}\mathbf{1}^T$, with $\zeta$ being a tunable parameter controlling strength of regularizations. It is also easy to see that in the trimming, $\hat{A}$ is set to be the adjacency matrix and $\hat{R}$ contains entries to remove columns or rows with high degrees from $A$.

For spectral algorithms using the non-backtracking matrix, its relation to form Eq. (1) is not straightforward. However we can link them using the theory of graph zeta function [8] which says that an eigenvalue $\mu$ of the non-backtracking operator satisfies the following quadratic eigenvalue equation,

$$\det[\mu^2 I - \mu A + (D - I)] = 0,$$

where $I$ is the identity matrix. It indicates that a particular vector $v$ that is related to the eigenvector of the non-backtracking matrix satisfies $(A - \frac{D-I}{\mu})v = \mu v$. Thus spectral clustering algorithm using the non-backtracking matrix is equivalent to the spectral clustering algorithm using matrix with form in Eq. (1), while $\hat{A} = A$, $\hat{R} = \frac{D-I}{\mu}$, and $\mu$ acting as a parameter. We note here that the parameter does not necessarily be an eigenevalue of the non-backtracking matrix. Actually a range of parameters work well in practice, like those estimated from the spin-glass transition of the system [24]. So we have related different approaches of resolving localizations of spectral algorithm in sparse graphs into the framework of regularization. Although this relation is in the context of community detection in networks, we think it is a general point-of-view, when the data matrix has a general form rather than a $\{0, 1\}$ matrix.

As we have argued in the introduction, above three ways of regularization work from case to case and have different problems, especially when system has noise. It means that in the framework of regularizations, the effective regularization matrix $\hat{R}$ added by these methods do not work in a general way and is not robust. In our understanding, the problem arises from the fact that in all these methods, the form of regularization is *fixed* for all kinds of data, regardless of different reasons for the localization. Thus one way to solve the problem would be looking for the regularizations that are specific for the given data, as a feature. In the following section we will introduce our method explicitly addressing how to learn such regularizations from localized eigenvectors of the data matrix.

## 3 Learning regularizations from localized eigenvectors

The reason that the informative eigenvectors are hidden in the bulk is that some random eigenvectors have large eigenvalues, due to the localization which represent the local structures of the system. In the complementary side, if these eigenvectors are not localized, they are supposed to have smaller eigenvalues than the informative ones which reveal the global structures of the graph. This is the main assumption that our idea is based on.

In this work we use the *Inverse Participation Ratio* (IPR), $I(v) = \sum_{i=1}^{n} v_i^4$, to quantify the amount of localization of a (normalized) eigenvector $v$. IPR has been used frequently in physics, for example for distinguishing the extended state from the localized state when applied on the wave function [3]. It is easy to check that $I(v)$ ranges from $\frac{1}{n}$ for vector $\{\frac{1}{\sqrt{n}}, \frac{1}{\sqrt{n}}, ..., \frac{1}{\sqrt{n}}\}$ to $1$ for vector $\{0, ..., 0, 1, 0, ..., 0\}$. That is, a larger $I(v)$ indicates more localization in vector $v$.

Our idea is to create a matrix $L_X$ with similar structures to $A$, but with non-localized leading eigenvectors. We call the resulting matrix *X-Laplacian*, and define it as $L_X = A + X$, where matrix $A$ is the data matrix (or its variant), and $X$ is learned using the procedure detailed below:

---

**Algorithm 1:** Regularization Learning

---

**Input**: Real symmetric matrix $A$, number of eigenvectors $q$, learning rate $\eta = O(1)$, threshold $\Delta$.
**Output**: X-Laplacian, $L_X$, whose leading eigenvectors reveal the global structures in $A$.

1. Set $X$ to be all-zero matrix.

2. Find set of eigenvectors $U = \{u_1, u_2, ..., u_q\}$ associated with the first $q$ largest eigenvalues (in algebra) of $L_X$.

3. Identify the eigenvector $v$ that has the largest inverse participation ratio among the $q$ eigenvectors in $U$. That is, find $v = \text{argmax}_{u \in U} I(u)$.

4. if $I(v) < \Delta$, return $L_X = A + X$; Otherwise, $\forall i, X_{ii} \leftarrow X_{ii} - \eta v_i^2$, then go to step 2.

---

We can see that the regularization matrix $X$ is a diagonal matrix, its diagonal entries are learned gradually from the most localized vector among the first several eigenvectors. The effect of $X$ is to penalize the localized eigenvectors, by suppressing down the eigenvalues associated with the localized eigenvectors. The learning will continue until all $q$ leading eigenvectors are delocalized, thus are supposed to correlate with the global structure rather than the local structures. As an example, we show the effect of $X$ to the spectrum in Fig. 1. In the left panel, we plot the spectrum of the adjacency matrix (i.e. before learning $X$) and the X-Laplacian (i.e. after learning $X$) of a sparse network generated by the stochastic block model with $q = 2$ groups. For the adjacency matrix in the left panel, localized eigenvectors have large eigenvalues and contribute a tail to the semicircle, covering the informative eigenvalue, leaving only one eigenvalue, which corresponds to the eigenvector that essentially sorts vertices according to their degree, out of the bulk. The spectral density of X-Laplacian is shown in the right panel of Fig. 1. We can see that the right corner of the continues part of the spectral density appearing in the spectrum of the adjacency matrix , is missing here. This is because due to the effect of $X$, the eigenvalues that are associated with localized eigenvectors in the adjacency matrix are pushed into the bulk, maintaining a gap between the edge of bulk and the informative eigenvalue (being pointed by the left red arrow in the figure).

The key procedure of the algorithm is the learning part in step 4, which updates diagonal terms of matrix $X$ using the most localized eigenvector $v$. Throughout the paper, by default we use learning rate $\eta = 10$ and threshold $\Delta = 5/n$. As $\eta = O(1)$ and $v_i^2 = O(1/n)$, we can treat the learned entries in each step, $\hat{L}$, as a perturbation to matrix $L_X$. After applying this perturbation, we anticipate that an eigenvalue of $L$ changes from $\lambda_i$ to $\lambda_i + \hat{\lambda}_i$, and an eigenvector changes from $u_i$ to $u_i + \hat{u}_i$. If we assume that matrix $L_X$ is not ill-conditioned, and the first few eigenvectors that we care about are distinct, then we have $\hat{\lambda}_i = u_i^T \hat{L} u_i$. Derivation of the above expression is straightforward, but for the completeness we put the derivations in the SI text. In our algorithm, $\hat{L}$ is a diagonal matrix with entries $\hat{L}_{ii} = -\eta v_i^2$ with $v$ denoting the identified eigenvector who has the largest inverse participation ratio, so last equation can be written as $\hat{\lambda}_i = -\eta \sum_k v_k^2 u_{ik}^2$. For the identified vector $v$, we further have

$$\hat{\lambda}_v = -\eta \sum_i v_i^4 = -\eta I(v). \tag{2}$$

It means the eigenvalue of the identified eigenvector with inverse participation ratio $I(v)$ is decreased by amount $\eta I(v)$. That is, *the more localized the eigenvector is, the larger penalty on its eigenvalue.*

In addition to the penalty to the localized eigenvalues, We see that the leading eigenvectors are delocalizing during learning. We have analyzed the change of eigenvectors after the perturbation given by the identified vector $v$, and obtained (see SI for the derivations) the change of an eigenvector $\hat{u}_i$ as a function of all the other eigenvalues and eigenvectors, $\hat{u}_i = \sum_{j \neq i} \frac{\sum_k u_{jk} v_k^2 u_{ik}}{\lambda_i - \lambda_j} u_j$. Then the inverse participation ratio of the new vector $u_i + \hat{u}_i$ can be written as

$$I(u_i + \hat{u}_i) = I(u_i) - 4\eta \sum_{l=1}^n \sum_{j \neq i} \frac{u_{jl}^2 v_l^2 u_{il}^4}{\lambda_i - \lambda_j} - 4\eta \sum_{l=1}^n \sum_{j \neq i} \sum_{k \neq l} \frac{u_{il}^3 v_k^2 u_{jk} u_{ik} u_{jl}}{\lambda_i - \lambda_j}. \tag{3}$$

As eigenvectors $u_i$ and $u_j$ are orthogonal to each other, the term $4\eta \sum_{l=1}^n \sum_{j \neq i} \frac{u_{jl}^2 v_l^2 u_{il}^4}{\lambda_i - \lambda_j}$ can be seen as a signal term and the last term can be seen as a cross-talk noise with zero mean. We see that the cross-talk noise has a small variance, and empirically its effect can be neglected. For the

leading eigenvector corresponding to the largest eigenvalue $\lambda_i = \lambda_1$, it is straightforward to see that the signal term is strictly positive. Thus if the learning is slow enough, the perturbation will always decrease the inverse participation ratio of the leading eigenvector. This is essentially an argument for convergence of the algorithm. For other top eigenvectors, i.e. the second and third eigenvectors and so on, though $\lambda_i - \lambda_j$ is not strictly positive, there are much more positive terms than negative terms in the sum, thus the signal should be positive with a high probability. Thus one can conclude that the process of learning $X$ makes first few eigenvectors de-localizing.

An example illustrating the process of the learning is shown in Fig. 2 where we plot the second eigenvector vs. the third eigenvector, at several times steps during the learning, for a network generated by the stochastic block model with $q = 3$ groups. We see that at $t = 0$, i.e. without learning, both eigenvectors are localized, with a large range of distribution in entries. The color of eigenvectors encodes the group membership in the planted partition. We see that at $t = 0$ three colors are mixed together indicating that two eigenvectors are not correlated with the planted partition. At $t = 4$ three colors begin to separate, and range of entry distribution become smaller, indicating that the localization is lighter. At $t = 25$, three colors are more separated, the partition obtained by applying k-means algorithm using these vectors successfully recovers $70\%$ of the group memberships. Moreover we can see that the range of entries of eigenvectors shrink to $[-0.06, 0.06]$, giving a small inverse participation ratio.

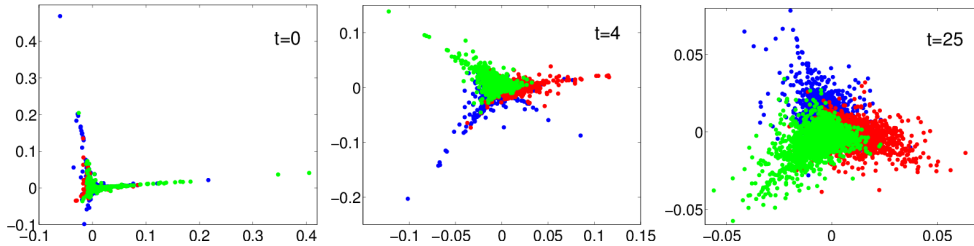

Figure 2: The second eigenvector $V_2$ compared with the third eigenvector $V_3$ of $L_X$ for a network at three steps with $t = 0, 4$ and $25$ during learning. The network has $n = 42000$ nodes, $q = 3$ groups, average degree $c = 3$, $\epsilon = 0.08$, three colors represent group labels in the planted partition.

## 4  Numerical evaluations

In this section we validate our approach with experiments on several inference problems, i.e. community detection problems, clustering from sparse pairwise entries, rank estimation and matrix completion from a few entries. We will compare performance of the X-Laplacian (using mean-removed data matrix) with recently proposed state-of-the-art spectral methods in the sparse regime.

### 4.1  Community Detection

First we use synthetic networks generated by the stochastic block model [9], and its variant with noise [10]. The standard Stochastic Block Model (SBM), also called the planted partition model, is a popular model to generate ensemble of networks with community structure. There are $q$ groups of nodes and a planted partition $\{t_i^*\} \in \{1, ..., q\}$. Edges are generated independently according to a $q \times q$ matrix $\{p_{ab}\}$. Without loss of generality here we discuss the commonly studied case where the $q$ groups have equal size and where $\{p_{ab}\}$ has only two distinct entries, $p_{ab} = c_{\text{in}}/n$ if $a = b$ and $c_{\text{out}}/n$ if $a \neq b$. Given the average degree of the graph, there is a so-called detectability transition $\epsilon^* = c_{\text{out}}/c_{\text{in}} = (\sqrt{c} - 1)/(\sqrt{c} - 1 + q)$ [7] , beyond which point it is not possible to obtain any information about the planted partition. It is also known spectral algorithms based on the non-backtracking matrix succeed all the way down to the transition [15]. This transition was recently established rigorously in the case of $q = 2$ [20, 21]. Comparisons of spectral methods using different matrices are shown in Fig. 3 left. From the figure we see that the X-Laplacian works as well as the non-backtracking matrix, down to the detectability transition. While the direct use of the adjacency matrix, i.e. $L_X$ before learning, does not work well when $\epsilon$ exceeds about $0.1$.

In the right panel of Fig. 3, each network is generated by the stochastic block model with the same parameter as in the left panel, but with 10 extra cliques, each of which contains 10 randomly selected

nodes. Theses cliques do not carry information about the planted partition, hence act as noise to the system. In addition to the non-backtracking matrix, X-Laplacian, and the adjacency matrix, we put into comparison the results obtained using other classic and newly proposed matrices, including Bethe Hessian [24], Normalized Laplacian (N. Laplacian) $L_{\text{sym}} = I - \tilde{A}$, and regularized and normalized Laplacian (R.N. Laplacian) $L_A = \tilde{A} - \zeta \mathbf{1}\mathbf{1}^{\mathbf{T}}$, with a optimized regularization $\zeta$ (we have scanned the whole range of $\zeta$, and chosen an optimal one that gives the largest overlap, i.e. fraction of correctly reconstructed labels, in most of cases). From the figure we see that with the noise added, only X-Laplacian works down to the original transition (of SBM without cliques). All other matrices fail in detecting the community structure with $\epsilon > 0.15$.

We have tested other kinds of noisy models, including the noisy stochastic block model, as proposed in [10]. Our results show that the X-Laplacian works well (see SI text) while all other spectral methods do not work at all on this dataset [10]. Moreover, in addition to the classic stochastic block model, we have extensively evaluated our method on networks generated by the degree-corrected stochastic block model [12], and the stochastic block model with extensive triangles. We basically obtained qualitatively results as in Fig. 3 that the X-Laplacian works as well as the state-of-the-art spectral methods for the dataset. The figures and detailed results can be found at the SI text.

We have also tested real-world networks with an expert division, and found that although the expert division is usually easy to detect by directly using the adjacency matrix, the X-Laplacian significantly improves the accuracy of detection. For example on the political blogs network [1], spectral clustering using the adjacency matrix gives 83 mis-classified labels among totally 1222 labels, while the X-Laplacian gives only 50 mis-classified labels.

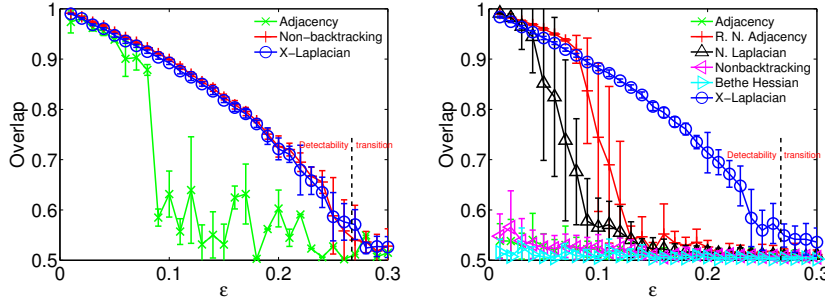

Figure 3: Accuracy of community detection, represented by overlap (fraction of correctly reconstructed labels) between inferred partition and the planted partition, for several methods on networks generated by the stochastic block model with average degree $c = 3$ (*left*) and with extra 10 size-10 cliques (*right*). All networks has $n = 10000$ nodes and $q = 2$ groups, $\epsilon = c_{\text{out}}/c_{\text{in}}$. The black dashed lines denote the theoretical detectability transition. Each data point is averaged over 20 realizations.

## 4.2 Clustering from sparse pairwise measurements

Consider the problem of grouping $n$ items into clusters based on the similarity matrix $S \in \mathbb{R}^{n \times n}$, where $S_{ij}$ is the pairwise similarity between items $i$ and $j$. Here we consider not using all pairwise similarities, but only $O(n)$ random samples of them. In other words, the similarity graph which encodes the information of the global clustering structure is sparse, rather than the complete graph. There are many motivations for choosing such sparse observations, for example in some cases all measurements are simply not available or even can not be stored.

In this section we use the generative model recently proposed in [26], since there is a theoretical limit that can be used to evaluate algorithms. Without loss of generality, we consider the problem with only $q = 2$ clusters. The model in [26] first assigns items hidden clusters $\{t_i\} \in \{1, 2\}^n$, then generates similarity between a randomly sampled pairs of items according to probability distribution, $p_{\text{in}}$ and $p_{\text{out}}$, associated with membership of two items. There is a theoretical limit $\hat{c}$ satisfying $\frac{1}{\hat{c}} = \frac{1}{q} \int ds \frac{(p_{\text{in}}(s) - p_{\text{out}}(s))^2}{p_{\text{in}}(s) + (q-1)p_{\text{out}}(s)}$, that with $c < \hat{c}$ no algorithm could obtain any partial information of the planted clusters; while with $c > \hat{c}$ some algorithms, e.g. spectral clustering using the Bethe Hessian [26], achieve partial recovery of the planted clusters.

Similar to the community detection in sparse graphs, spectral algorithms directly using the eigenvectors of a similarity matrix $S$ does not work well, due to the localization of eigenvectors induced by the sparsity. To evaluate whether our method, the X-Laplacian, solves the localization problem, and how it works compared with the Bethe Hessian, in Fig. 4 we plot the performance (in overlap, the fraction of correctly reconstructed group labels) of three algorithms on the same set of similarity matrices. For all the datasets there are two groups with distributions $p_{in}$ and $p_{out}$ being Gaussian with unit variance and mean $0.75$ and $-0.75$ respectively. In the left panel of Fig. 4 the topology of pairwise entries is random graph, Bethe Hessian works down to the theoretical limit, while directly using of the measurement matrix gives a poor performance. We can also see that X-Laplacian has fixed the localization problem of directly using of the measurement matrix, and works almost as good as the Bethe-Hessian. We note that the Bethe Hessian needs to know the parameters (i.e. parameters of distributions $p_{in}$ and $p_{out}$), while the X-Laplacian does not use them at all.

In the right panel of Fig. 4, on top of the ER random graph topology, we add some noisy local structures by randomly selecting 20 nodes and connecting neighbors of each selected node to each other. The weights for the local pairwise were set to 1, so that the noisy structures do not contain information about the underlying clustering. We can see that Bethe Hessian is influenced by noisy local structures and fails to work, while X-Laplacian solves the localization problems induced by sparsity, and is robust to the noise. We have also tested other kinds of noise by adding cliques, or hubs, and obtained similar results (see SI text).

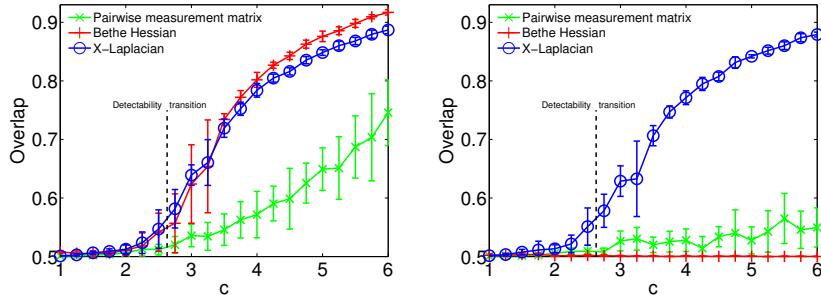

Figure 4: Spectral clustering using sparse pairwise measurements. The X-axis denotes the average number of pairwise measurements per data point, and the Y-axis is the fraction of correctly reconstructed labels, maximized over permutations. The model used to generate pairwise measurements is proposed in [26], see text for detailed descriptions. In the left panel, the topologies of the pairwise measurements are random graphs. In the right panel in addition to the random graph topology there are 20 randomly selected nodes with all their neighbors connected. Each point in the figure is averaged over 20 realizations of size $10^4$.

## 4.3 Rank estimation and Matrix Completion

The last problem we consider in this paper for evaluating the X-Laplacian is completion of a low rank matrix from few entries. This problem has many applications including the famous collaborative filtering. A problem that is closely related to it is the rank estimation from revealed entries. Indeed estimating rank of the matrix is usually the first step before actually doing the matrix completion. The problem is defined as follows: let $A^{\text{true}} = UV^T$, where $U \in \mathbb{R}^{n \times r}$ and $V \in \mathbb{R}^{m \times r}$ are chosen uniformly at random and $r \ll \sqrt{nm}$ is the ground-true rank. Only few, say $c\sqrt{mn}$, entries of matrix $A^{\text{true}}$ are revealed. That is we are given a matrix $A \in \mathbb{R}^{n \times m}$ who contains only subset of $A^{\text{true}}$, with other elements being zero. Many algorithms have been proposed for matrix completion, including nuclear norm minimization [5] and methods based on the singular value decomposition [4] etc. Trimming which sets to zero all rows and columns with a large revealed entries, is usually introduced to control the localizations of singular vectors and to estimate the rank using the gap of singular values [14]. Analogous to the community detection problem, trimming is not supposed to work optimally when matrix $A$ is sparse. Indeed in [25] authors reported that their approach based on the Bethe Hessian outperforms trimming+SVD when the topology of revealed entries is a sparse random graph. Moreover, authors in [25] show that the number of negative eigenvalues of the Bethe Hessian gives a more accurate estimate of the rank of $A$ than that based on trimming+SVD.

However, we see that if the topology is not locally-tree-like but with some noise, for example with some additional cliques, both trimming of the data matrix and Bethe Hessian perform much worse, reporting a wrong rank, and giving a large reconstruction error, as illustrated in Fig. 5. In the left panel of the figure we plot the eigenvalues of the Bethe Hessian, and singular values of trimmed matrix $A$ with true rank $r^{\text{true}} = 2$. We can see that both of them are continuously distributed: there is no clear gap in singular values of trimmed $A$, and Bethe Hessian has lots of negative eigenvalues. In this case since matrix $A$ could be a non-squared matrix, we need to define the X-Laplacian as

$L_X = \begin{pmatrix} 0 & A \\ A & 0 \end{pmatrix} - X$. The eigenvalues of $L_X$ are also plotted in Fig. 5 where one can see clearly

that there is a gap between the second largest eigenvalue and the third one. Thus the correct rank can be estimated using the value minimizing consecutive eigenvalues, as suggested in [14].

After estimating the rank of the matrix, matrix completion is done by using a local optimization algorithm [27] starting from initial matrices, that obtained using first $r$ singular vectors of trimming+SVD, first $r$ eigenvectors of Bethe Hessian and X-Laplacian with estimated rank $r$ respectively. The results are shown in Fig. 5 right where we plot the probability that obtained root mean square error (RMSE) is smaller than $10^{-7}$ as a function of average number of revealed entries per row $c$, for the ER random-graph topology plus noise represented by several cliques. We can see that X-Laplacian outperforms Bethe Hessian and Trimming+SVD with $c \geq 13$. Moreover, when $c \geq 18$, for all instances, only X-Laplacian gives an accurate completion for all instances.

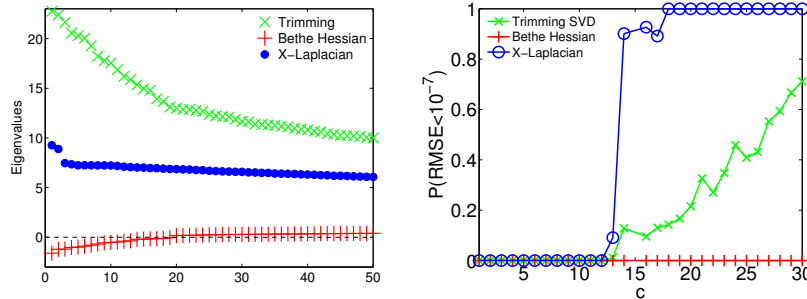

Figure 5: *(Left:)* Singular values of sparse data matrix with trimming, eigenvalues of the Bethe Hessian and X-Laplacian. The data matrix is the outer product of two vectors of size 1000. Their entries are Gaussian random variables with mean zero and unit variance, so the rank of the original matrix is 2. The topology of revealed observations are random graphs with average degree $c = 8$ plus 10 random cliques of size 20. *(Right:)* Fraction of samples that RMSE is smaller than $10^{-7}$, among 100 samples of rank-3 data matrix $UV^T$ of size $1000 \times 1000$, with the entries of $U$ and $V$ drawn from a Gaussian distribution of mean 0 and unit variance. The topology of revealed entries is the random graph with varying average degree $c$ plus 10 size-20 cliques.

## 5   Conclusion and discussion

We have presented the X-Laplacian, a general approach for detecting latent global structure in a given data matrix. It is completely a data-driven approach that learns different forms of regularization for different data, to solve the problem of localization of eigenvectors or singular vectors. The mechanics for de-localizing of eigenvectors during learning of regularizations has been illustrated using the matrix perturbation analysis. We have validated our method using extensive numerical experiments, and shown that it outperforms state-of-the-art algorithms on various inference problems in the sparse regime and with noise.

In this paper we discuss the X-Laplacian using directly the (mean-removed) data matrix $A$, but we note that the data matrix is not the only choice for the X-Laplacian. Actually we have tested approaches using various variants of A, such as normalized data matrix $\tilde{A}$, and found they work as well. We also tried learning regularizations for the Bethe Hessian, and found it succeeds in repairing Bethe Hessian when Bethe Hessian has localization problem. These indicate that our scheme of regularization-learning is a general spectral approach for hard inference problems.

A (Matlab) demo of our method can be found at http://panzhang.net.

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
