[Supplementary Material]

# Supplementary Information for:
# Robust Spectral Detection of Global Structures in the Data by Learning a Regularization

**Pan Zhang**

Institute of Theoretical Physics, Chinese Academy of Sciences, Beijing 100190, China
panzhang@itp.ac.cn

## 1 Perturbation analysis

After applying the perturbation, We anticipate that an eigenvalue of $L_X$ changes from $\lambda_i$ to $\lambda_i + \hat{\lambda}_i$, and an eigenvector changes from $u_i$ to $u_i + \hat{u}_i$. If we assume that matrix $L_X$ is not ill-conditioned, and the first few eigenvectors that we care about are distinct, then we have

$$(L_X + \hat{L})(u_i + \hat{u}_i) = (\lambda_i + \hat{\lambda}_i)(u_i + \hat{u}_i).$$

By making use of

$$L_X u_i = \lambda_i u_i,$$

and keeping only first order terms, we have

$$\hat{L}u_i + L_X \hat{u}_i = \lambda_i \hat{u}_i + \hat{\lambda}_i u_i. \tag{1}$$

Since $L_X$ is a real symmetric matrix, we can represent $\hat{u}_i$ as a weighted sum of eigenvectors of $L_X$, as

$$\hat{u}_i = \sum_{j=1}^{n} \omega_j u_j, \tag{2}$$

where $\omega_j$ is the coefficient and $u_j$ is j'th eigenvector of $L_X$. Insert last equation into Eq. (1), we have

$$\hat{L}u_i + L_X \sum_{j=1}^{n} \omega_j u_j = \lambda_i \sum_{j=1}^{n} \omega_j u_j + \hat{\lambda}_i u_i, \tag{3}$$

which evaluates to

$$\hat{L}u_i + \sum_{j=1}^{n} \omega_j \lambda_j u_j = \lambda_i \sum_{j=1}^{n} \omega_j u_j + \hat{\lambda}_i u_i. \tag{4}$$

Multiplying $u_i^T$ to both sides of last equation results to

$$u_i^T \hat{L}u_i + \sum_{j=1}^{n} \omega_j \lambda_j u_i^T u_j = \lambda_i \sum_{j=1}^{n} u_i^T \omega_j u_j + \hat{\lambda}_i u_i^T u_i. \tag{5}$$

Notice that in the last equation $u_i^T u_i = 1$ and the second term in the left hand side and the first term in the right hand side cancel each other, thus we have

$$\hat{\lambda}_i = u_i^T \hat{L}u_i. \tag{6}$$

In our algorithm, $\hat{L}$ is a diagonal matrix with entries $\hat{L}_{ii} = -\eta v_i^2$ where $v_i$ denotes the $i$'th element of the selected eigenvector $v$ who has the largest inverse participation ratio. Thus the shift of an eigenvalue $\lambda_j$ associated with eigenvector $u_j$ (which is different from $v$) is then

$$\hat{\lambda}_j = -\eta \sum_{i=1}^{n} v_i^2 u_{ji}^2. \tag{7}$$

For the selected vector $v$, the change of its eigenvalue is

$$\hat{\lambda}_v = -\eta \sum_{i=1}^{n} v_i^4 = -\eta I(v). \tag{8}$$

That is, the amount of decreasing of eigenvalue associated with the selected vector is proportional to its inverse participation ratio.

In addition to the shift of eigenvalues, we can also derive the change of eigenvectors after perturbation. Multiplying transpose of an eigenvector $u_j$ to both sides of Eq.(4) results to

$$u_j^T \hat{L} u_i + \sum_{k=1}^{n} \omega_k \lambda_k u_j^T u_k = \lambda_i \sum_{k=1}^{n} u_j^T \omega_k u_k, \tag{9}$$

which evaluates to

$$u_j^T \hat{L} u_i + \omega_j \lambda_j = \lambda_i \omega_j, \tag{10}$$

where we can find that

$$\omega_j = \frac{u_j^T \hat{L} u_i}{\lambda_i - \lambda_j}. \tag{11}$$

Given that the perturbation is $\hat{L}_{ii} = -\eta v_i^2$, we have an expression for the change of an eigenvector

$$
\begin{aligned}
\hat{u}_i &= \sum_{j \neq i} \frac{u_j^T \hat{L} u_i}{\lambda_i - \lambda_j} u_j \\
&= -\eta \sum_{j \neq i} \frac{\sum_k u_{jk} v_k^2 u_{ik}}{\lambda_i - \lambda_j} u_j.
\end{aligned} \tag{12}
$$

Notice that the inverse participate ratio of the new vector $u_i + \hat{u}_i$ is

$$I(u_i + \hat{u}_i) = \sum_{l=1}^{n} (u_{il} + \hat{u}_{il})^4, \tag{13}$$

Expand above equation to the first order of $\hat{u}_{il}$, we have

$$
\begin{aligned}
I(u_i + \hat{u}_i) &\approx I(u_i) + 4 \sum_{l=1}^{n} u_{il}^3 \hat{u}_{il} \\
&= I(u_i) - 4\eta \sum_{l=1}^{n} u_{il}^3 \sum_{j \neq i} \frac{\sum_k u_{jk} v_k^2 u_{ik}}{\lambda_i - \lambda_j} u_{jl} \\
&= I(u_i) - 4\eta \sum_{l=1}^{n} \sum_{j \neq i} \frac{u_{jl}^2 v_l^2 u_{il}^4}{\lambda_i - \lambda_j} - 4\eta \sum_{l=1}^{n} \sum_{j \neq i} \sum_{k \neq l} \frac{u_{il}^3 v_k^2 u_{jk} u_{ik} u_{jl}}{\lambda_i - \lambda_j}
\end{aligned} \tag{14}
$$

## 2   Detailed process of learning a regularization

In Fig. 2 we plot the evolution of eigenvalues, overlap and the Inverse Participation Ratio (IPR) for the second, third and forth eigenvectors during learning of the X-Laplacian for a network generated by the stochastic block model. The network has a community structure with 3 groups, however the first three eigenvectors of the adjacency matrix are localized (see left panel at $t = 0$) and do not reveal the underlying community structure (see the right panel at $t$ small. We can also see from the left panel that the IPR of them are decreasing as $t$ increases during learning. From the middle panel of the figure, we see that all the 3 eigenvalues are decreasing, while the spectral gap $D_3 - D_4$ is increasing during learning. It is interesting to see that at $t = 4$, there is a exchange of positions of the third eigenvector and the forth eigenvector. This gives a bump of the IPR, as well as an increase of accuracy of detection (characterized by overlap) at $t = 4$.

Figure 1: Inverse Participation Ratio of first three eigenvectors ($I_1, I_2, I_3$, overlap (the fraction of correctly reconstructed labels) and first three eigenvalues ($D_1, D_2, D_3$) as a function of learning steps $t$ for a network generated by the stochastic block model with $n = 42000$, $q = 3$ groups, average degree $c = 3$, $\epsilon = 0.08$ and learning rate $\eta = 1$. The overlap is the fraction of successfully reconstructed labels, maximized over group permutations.

## 3    Additional numerical evaluations on community detections

Here we compare the performance of the X-Laplacian with other state-of-art spectral algorithms on variants of the stochastic block model, namely the degree-corrected stochastic block model [2] and the triangular stochastic block model [4] which is the stochastic block model with triangles. It is known that in the stochastic block model, there is a detectability transition at

$$\epsilon^* = (\sqrt{\hat{c}} - 1)/(\sqrt{\hat{c}} - 1 + q),$$

where $\hat{c}$ is the excess average degree

$$\hat{c} = \frac{\langle k^2 \rangle}{\langle k \rangle} - 1,$$

and the spectral clustering algorithm based on the non-backtracking matrix achieves this threshold. In the left panel of Fig.2 we compare the performance (evaluated using the overlap, fraction of correctly reconstructed labels) of spectral algorithms using the adjacency matrix, the non-backtracking matrix and the X-Laplacian on networks generated by the degree corrected stochastic block model with a power-law degree distribution with exponent $-2.5$. As the figure shows, our approach works even better than the algorithm using the non-backtracking matrix, this is because when the networks size ($10^4$) is not large enough, the long tails of degree distribution creates short loops in the network, downgrading the performance of the algorithm using the non-backtracking matrix which is supposed to work optimally in the locally-tree like networks.

For the triangular stochastic block model, due to the presence of triangles, the non-backtracking matrix suffers from short loops and does not work well. In this case the generalized non-backtracking matrix, which runs on a factor graph with both edges and triangles treated as function nodes, works down to the transition [4]. In the right panel of Fig. 2 we compare the performance of the spectral algorithm using the adjacency matrix, generalized non-backtracking matrix and the X-Laplacian , and we can see that X-Laplacian works as well as the generalized non-backtracking matrix and down to the transition.

It has been reported in  [1] that on the perturbed stochastic block model, spectral algorithms including the one using Bethe Hessian do fail in detecting the community structures, while other method, e.g. semi-definite programming, works well. In the perturbed stochastic block model, after a network is generated by the stochastic block model, neighbors of some randomly selected nodes are connected to each other acting as noise to the underlying community structure, In Fig. 3 we numerically examined the performance of X-Laplacian using $\tilde{A}$, i.e. $L_X = \tilde{A} + X$, on networks generated by perturbed stochastic block model  [1], with exactly the same network size and parameters as in [1] (see Fig. 5 in their SI), with parameter $a$ and $b$ denoting expected number of edges per node connecting nodes in the same group and in different groups, respectively. By comparing Fig. 3 with Fig. 5 in SI of [1] we can see from X-Laplacian works similarly to the semi-definite programming while Bethe-Hessian based method does not work at all.

Figure 2: Accuracy of community detection, represented by overlap between inferred partition and the planted partition, for several method on networks generated by the degree corrected stochastic block model, and a power-law degree distribution with exponent $-2.5$ (*left*); triangular stochastic block model (*right*) with average degree $c = 3$ and $\rho = 0.5$ which means half of edges belong to triangles rather than single edges [unpublished]. All networks has $n = 10000$ nodes and $q = 2$ groups. In $X$-axis, $\epsilon = c_{\text{out}}/c_{\text{in}}$ controls the hardness of the problem. Each data point is averaged over 20 realizations.

Figure 3: X-Laplacian using $\tilde{A}$ on networks generated by the perturbed stochastic block model [1], with exactly the same network size and parameters as in Fig.5 of SI in [1]. $a$ and $b$ denote average degree connecting nodes in the same group and different groups respectively, and $\alpha$ is the fraction of selected noisy nodes. Each point is averaged over 20 instances.

# 4    Additional numerical evaluations on spectral clustering using pairwise similarity measurements

In this section we compare the performance of spectral algorithms using the data matrix, the Bethe Hessian and the X-Laplacian, on the model recently proposed in [3], which generates pairwise measurements between two groups of nodes from different probability distributions. Two distributions $p_{\text{in}}$ and $p_{\text{out}}$ are chosen to be Gaussian with unit variance and mean $0.75$ and $-0.75$ respectively. On top of the network we add two different kinds of noise, i.e. cliques and hubs to the random graph topology. And from figures we can see that the results are qualitatively similar to right panel of Fig. 4 in the main text where X-Laplacian outperforms both Bethe Hessian and X-Laplacian in reconstructing the planted partition.

Figure 4: Spectral clustering using sparse pairwise measurements, using model proposed in [3]. The overlap in Y-axis is the fraction of correctly reconstructed labels, X-axis denotes the average number of pairwise measurements per data point. (*Left*): the topologies are random graphs with average degree $c$ together with 10 size-20 cliques. (*Right*): panel the topologies are random graphs with average degree $c$ together with 10 hubs whose degrees are 50. Each point in the figure is averaged over 20 realizations of data set of size $10^4$.