[Reviews · NeurIPS 2016]

Reviewer 1

Summary

The paper proposes a new type of regularized laplacian for performing community detection, clustering and matrix completion. The paper proposes a way to learn the regularization of the laplacian based on the network data, such that community detection and such tasks are possible for sparse networks. The new type of regularization has also been claimed to produced a more robust laplacian, which uncovers cluster structure even under errors in network data.

Qualitative Assessment

The paper proposes a way to learn the regularization of laplacians for dealing with the localization of eigenvectors of sparse and noisy networks. The iterative method of learning the regularization that has been proposed in the paper seems promising. However, the technical portion of the paper is lacking a bit. The paper proposes a new quantity - Inverse Participation ratio (IPR), but does not link the IPR with community detection performances theoretically. The paper claims that lower the IPR the better the eigenvector is for sparse and noisy networks, however, no formal result is stated about the bounds of IPR of the regularized laplacians, under which community detection will be possible. Also, the formal proof of effectiveness of X-laplacian in recovering community is not provided (which has been recently proved for regularized laplacian and non-backtracking matrices). No formal definition of robustness of the laplacians and how X-laplacians achieve them has been stated and proved in the paper. The paper uses simulation to substantiate many of the claims. The paper tries to build a nice connection between the different forms of regularized laplacian/adjacency matrices proposed in the literature for sparse networks. The way of learning the regularization is also nice. The X-laplacian method, if properly substantiated, can be quite a good way of achieving community detection and clustering for sparse and noisy networks. The writing of the paper is okay. It has some typos (like, pg 2, line 68, by learn should be by learning; pg 3, line 116, X-Lapacian should be X-Laplacian). The notation epsilon and epsilon* has been interchangingly used in page 5 and 6. Overall, the paper seems a bit pre-mature. It contains a nice idea, but a more formal analysis of X-Laplacians should be done before it is ready for publication.

Confidence in this Review

2-Confident (read it all; understood it all reasonably well)


Reviewer 2

Summary

This paper proposes, in spectral detection of global structures in data matrix, a means to resolve the localization problem, which is known to cause several spectral detection methods to fail in sparse and/or noisy cases. The proposal is based on learning a proper regularizer on the basis of localized eigenvectors of the data matrix, where the algorithm is such that one iteratively updates the regularizer so that the regularized matrix has no significant eigenvalues, with the associated eigenvectors being localized in terms of the inverse participation ratio (IPR).

Qualitative Assessment

Although the use of IPR in quantifying localization would work well in practice, as demonstrated in the section on numerical evaluations, it seems heuristic, since there would be several different quantities which would have qualitatively the same property as the IPR. Why it is a good quantifier compared with other possibilities in terms of detection of global structures should be discussed. The author claims that the approach of trimming throws away part of the information, but one can see in the algorithm proposed in [12] that while trimming is used in the first stage the whole dataset is used in the final stage. I was not convinced by the argument in page 3, which claims that spectral algorithms using the non-backtracking matrix can also be interpreted as an example of equation (1). In the interpretation the regularizer is dependent on the eigenvalue, which obscures the interpretation of these spectral algorithms as instances of the regularized spectral method (equation (1)). Specifically, I do not see any description of the claimed direct relation between the vector v appearing in Line 90 and the eigenvectors of the non-backtracking matrix. Minor points: - Line 32: Weigner's -> Wigner's - Line 35: the wigner's -> Wigner's - Line 37: which (is -> are) roughly - Line 68: by learn(ing) a proper - Line 75: that correlate(d) - Line 103: that (are) specific - Line 108: have (a large eigenvalue -> large eigenvalues) / which represent(s) - Line 110: one who reveal -> ones which reveal - Line 127: semi-cycle -> semi-circle (But in this case the bulk of the spectrum is not exactly a semi-circle.) - Line 129: The spectra(l) density - Line 130: the continu(es -> ous) part - Line 152: the inverse participat(e -> ion) ratio - Lines 179, 293: state-of-art -> state-of-the-art - Line 220: between (duplicate) - Line 227: assign items a hidden clusters -> assigns to items hidden clusters - Line 228: then generate(s) similarity between (a) randomly sampled pairs of items - Line 236: Here the term overlap is briefly explained, but it has already appeared several times before. The brief explanation should be given at its first appearance. - Line 246: each selected node(s) - Line 249: fail(s) to work - Line 255: that (is) closely related - Line 276: where (one) can see - Line 286: a(n) accurate completion

Confidence in this Review

2-Confident (read it all; understood it all reasonably well)


Reviewer 3

Summary

This paper proposes a self-adaptive regularization method for spectral clustering. Various versions of spectral clustering from sparse data (sparse networks, or sparse set of similarities) all share the undesirable property that there are small structures in the graphs (either naturally, e.g. high degree hubs or hanging trees, or put by an adversary) that create large eigenvalues in the spectrum with eigen-vectors localized on the structure. Several methods (discussed in the paper) were previously suggested to deal with this problem, but none of them seems to be very universal. The current paper proposed an adaptive way to learn a regularization of the Laplacian, called x-Laplacian, and illustrated that this strategy makes the spectral clustering robust to a wide range of perturbations.

Qualitative Assessment

I find this paper very exciting. It is possibly a generic way how to make spectral methods robust to small adversarial changes in the output in a similar manner as e.g. semi-definite programming is known to be. The proposed method is clear and relatively simple to understand and implement. The present paper provides very good empirical assessment of the method. Not so much on the theoretical side, but I think this will definitely trigger research in this direction (making the implementation more efficient, providing guarantees, etc.). However, the language and grammar are not very good and surely the authors should make additional effort in this direction (out of many Wigner is not "Weigner" nor "wigner", methods are popular as *they give*, ...). The words "detectability transition" are unreadable in Fig. 3, and barely in Fig. 4.

Confidence in this Review

3-Expert (read the paper in detail, know the area, quite certain of my opinion)


Reviewer 4

Summary

A spectral algorithm for the detection of global structures in data is presented. Motivations and previous approaches are discussed. The main contribution is an adaptive regularizer which penalizes localized eigenvectors.

Qualitative Assessment

The proposed method is a bit ad-hoc but seems to have some positive effects. For me it wasn't quite clear what Figure 1 is supposed to illustrate.

Confidence in this Review

2-Confident (read it all; understood it all reasonably well)


Reviewer 5

Summary

This paper presents a method to regularize sparse matrix spectral analysis problems. The authors claim to solve the localized eigenvectors' problem. A heuristic algorithm is proposed to learn the regularization which basically suppresses down eigenvalues associated with singular eigenvectors.

Qualitative Assessment

This paper proposes an algorithm to learn regularization to solve the problem of singular eigenvectors in sparse matrices. However, the goal of this paper is vague: the authors state that the goal is to create a non-localized leading eigenvectors matrix that is close to the original sparse matrix. Two things are unclear: 1. What does close to mean exactly? Is there a distance metric defined? Does the algorithm find the closest one or sub-optimal one? What about the perturbation error? 2. What does non-localized leading eigenvectors matrix mean? Do you assume full rank? What if the sparse matrix is of high dimension, but the rank is actually low? I believe that the idea behind the algorithm is neat, but I would be more convinced if the authors provided more precise theoretical analysis or real world applications.

Confidence in this Review

2-Confident (read it all; understood it all reasonably well)


Reviewer 6

Summary

This paper discussed regularization as a united framework for community detection problem in sparse graphs, and proposed an algorithm to learn regularization from localized eigenvectors, through a proposed ratio: inverse participation ratio, which quantify the amount of localization of a (normalized) eigenvector. The algorithm works in sparse and noisy regime. They conducted experiments on community detection, clustering from sparse pairwise measurements, rank estimation and matrix completion.

Qualitative Assessment

The idea of this paper is novel and the analysis of how the authors come up with the algorithm is neat, they proposed an algorithm to learn regularizations form localized eigenvectors and it works in sparse and noisy regime, where spectral algorithms usually does not perform well. Plenty of experiments have been conducted to show the performance and I see this kind of method promising. However, the result seems preliminary as there is no proved guarantee of the algorithm. Some major problems are: (1) The analysis of this paper is just how to come up with the algorithm, they did a great job on explaining how to incorporate the proposed ratio in the algorithm using perturbation theory, but lack the proof on convergence analysis (i.e., how to choose \Delta to guarantee a successful recovery) and consistency analysis (i.e., how large is the recovery error). (2) Also, it is not clear how they come up with the ratio as it is a sum of 4th order elements, it seems to come from the design of the algorithm, but an analysis on whether this can be a generalized metric would be more helpful. Some minor problems/typos: (1) The authors use 'overlap' as accuracy for prediction, which is confusing as there is overlapping community detection problem (2) Equation 3 should begin with \approx not equal, as shown in SI (3) Figure 1 will be better if plotting curve of the density, rather than bar graph (4) 'I.e.' in 118 etc. is strange, conventionally 'i.e.' is used (5) Line 220, 'the pairwise similarity between between items' should be 'the pairwise similarity between items' (6) Line 226, 'Without loose of generality' should be 'Without loss of generality' (7) Line 227, 'assign items a hidden clusters' should be 'assign items hidden clusters' (8) Line 279, 'After the estimating the rank of the matrix' should be 'After estimating the rank of the matrix' (9) Figure 1 in SI, we have no idea what I2, I3, I4, D2, D3, D4 represent

Confidence in this Review

2-Confident (read it all; understood it all reasonably well)